# *Corynebacterium ulcerans* Infections in Eurasian Beavers (*Castor fiber*)

**DOI:** 10.3390/pathogens12080979

**Published:** 2023-07-26

**Authors:** Reinhard Sting, Catharina Pölzelbauer, Tobias Eisenberg, Rebecca Bonke, Birgit Blazey, Martin Peters, Karin Riße, Andreas Sing, Anja Berger, Alexandra Dangel, Jörg Rau

**Affiliations:** 1Chemical and Veterinary Analysis Agency (CVUA) Stuttgart, 70736 Fellbach, Germany; catharina.poelzelbauer@cvuas.bwl.de (C.P.); birgit.blazey@cvuas.bwl.de (B.B.); joerg.rau@cvuas.bwl.de (J.R.); 2Consiliary Laboratory for Corynebacterium pseudotuberculosis (DVG), 70736 Fellbach, Germany; 3Hessian State Laboratory (LHL), 35392 Giessen, Germany; tobias.eisenberg@lhl.hessen.de (T.E.); rebecca.bonke@lhl.hessen.de (R.B.); karin.risse@lhl.hessen.de (K.R.); 4Chemical and Veterinary Investigation Office Westfalen, 59821 Arnsberg, Germany; martin.peters@cvua-westfalen.de; 5Germany National Consiliary Laboratory for Diphtheria, 85764 Oberschleißheim, Germany; andreas.sing@lgl.bayern.de (A.S.); anja.berger@lgl.bayern.de (A.B.); 6Bavarian Health and Food Safety Authority, 85764 Oberschleißheim, Germany; alexandra.dangel@lgl.bayern.de

**Keywords:** *Corynebacterium ulcerans*, diphtheria, zoonosis, abscess, *Streptococcus castoreus*, 16-23S ITS, *rpoB*, whole-genome sequencing

## Abstract

The Eurasian beaver (*Castor fiber)* has been reintroduced successfully in Germany since the 1990s. Since wildlife is an important source of zoonotic infectious diseases, monitoring of invasive and reintroduced species is crucial with respect to the One Health approach. Three Eurasian beavers were found dead in the German federal states of Bavaria, North Rhine–Westphalia and Baden–Wuerttemberg in 2015, 2021 and 2022, respectively. During post-mortem examinations, *Corynebacterium* (*C.*) *ulcerans* could be isolated from the abscesses of two beavers and from the lungs of one of the animals. Identification of the bacterial isolates at the species level was carried out by spectroscopic analysis using MALDI-TOF MS, FT-IR and biochemical profiles and were verified by molecular analysis based on 16-23S internal transcribed spacer (ITS) region sequencing. Molecular characterization of the *C. ulcerans* isolates using whole-genome sequencing (WGS) revealed a genome size of about 2.5 Mbp and a GC content of 53.4%. Multilocus sequence typing (MLST) analysis classified all three isolates as the sequence type ST-332. A minimum spanning tree (MST) based on cgMLST allelic profiles, including 1211 core genes of the sequenced *C. ulcerans* isolates, showed that the beaver-derived isolates clearly group on the branch of *C. ulcerans* with the closest relationship to each other, in close similarity to an isolate from a dog. Antibiotic susceptibility testing revealed resistance to clindamycin and, in one strain, to erythromycin according to EUCAST, while all isolates were susceptible to the other antimicrobials tested.

## 1. Introduction

The Eurasian beaver (*Castor fiber*) was a missing link in wildlife ecosystems due to its almost complete extinction to a population size of only about 1200 animals in Europe at the beginning of the 20th century [1]. This situation has changed fundamentally since the successful reintroduction of beavers throughout Europe in the early 1990s [2,3,4]. This positive development has led to beavers being assigned to category “Least Concern” in the IUCN Red List of Threatened Species [5]. However, the reintroduction of the Eurasian beaver in European countries bears the risks of importation and emergence of wildlife diseases and zoonoses, with the possible consequence of creation and establishment of a novel wildlife pathogen reservoir [6]. In this context, it should be considered that about 75% of known re-emerging and emerging human diseases originate from animal reservoirs [7,8,9]. The potential impact of re-emerging and emerging pathogens originating in free-ranging beavers with a resulting health hazard for both animals and humans cannot yet be fully assessed. Risk analysis and assessment of diseases emerging from pathogens in humans, domestic animals or other wildlife through the reintroduction of beavers into the wild have been conducted [10,11,12]. Thus, a small number of studies is available that present data on the occurrence of bacterial pathogens such as *Francisella tularensis* [13], extended-spectrum beta lactamase (ESBL) *E. coli* [6], *Salmonella* spp. [14], *Klebsiella pneumoniae* [15], *Yersinia enterocolitica* [16], *Campylobacter* [11] and *Leptospira* spp. [17], enterococci bearing virulence factors and vancomycin resistance [18], and pathogenic *Staphylococcus aureus* [19]. With an increasing beaver population density, it is conceivable that beavers will be hunted, which is already being seen during regular hunting activity in Northern and Eastern Europe [20]. This brings humans and other animals into more intensive contact with beavers. Therefore, wildlife disease surveillance and reports on the detection of pathogens, especially zoonotic agents in wildlife, are of paramount interest for the One Health approach.

Among zoonotic pathogens, *Corynebacterium* spp. belonging to the diphtheria species complex as well as new species with the potential of producing diphtheria toxin have attracted special attention [21]. In the last decade, the number of agents belonging to the diphtheria species complex has increased. This complex currently includes the human pathogenic bacteria *C. diphtheriae* and *C. belfantii*, the zoonotic agents *C. ulcerans*, *C. pseudotuberculosis* and presumably *C. rouxii* [22], and *C. silvaticum* [23]. Interestingly, the corynebacteria originating from humans and animals cluster phylogenetically together in Group Q of a protein-based phylogenetic tree [24], suggesting that other isolates from animals may be pathogenic to humans.

*C. ulcerans* occupies a special position within the corynebacteria because of its wide spectrum of hosts, including livestock, companion and wild animals, and humans [25,26]. Furthermore, *C. ulcerans* is characterized by variability in toxin production, including non-toxigenic (DT-negative) strains, the diphtheria *tox* gene-bearing and expressing strains (DT-positive), and non-toxigenic *tox*-bearing (NTTB) strains [21]. Of particular importance is the fact that *C. ulcerans* has evolved into the most important zoonotic *Corynebacterium* sp. causing diphtheria-like illness in Europe, superseding *C. diphtheriae* over the last two decades [26,27,28,29,30].

In this study, we present three temporally and geographically independent cases of infections with pathogenic *C. ulcerans* in Eurasian beavers in Germany for the first time. These case reports are of special relevance, because the *C. ulcerans* isolates can lead to fatal infections in beavers and render those isolates as potential zoonotic agents.

## 2. Materials and Methods

### 2.1. Examined Beavers

In general, all beavers that are found dead are sent to post-mortem examination for a passive wildlife health monitoring. An active health monitoring of beavers is not possible because beavers are strictly protected species and hunting, capturing and taking beavers from their natural habitat is strictly prohibited.

#### 2.1.1. Beaver 1

The first beaver (Beaver 1) was found dead in the county of Main–Spessart in the federal state of Bavaria, Germany on 21 September 2015 and was submitted to LHL Giessen for post-mortem and microbiological examinations on the same day.

#### 2.1.2. Beaver 2

The second beaver (Beaver 2) was found moribund at Wilhelmsruh on the Hevel in the county of Soest in the federal state of North Rhine–Westphalia, Germany on 7 April 2021. It was submitted to a local veterinarian and died two days later in the veterinary practice. Post-mortem and microbiological examinations were carried out at CVUA Westfalen in Arnsberg on 9 April 2021.

#### 2.1.3. Beaver 3

The third beaver (Beaver 3) was detected dead in the urban district of Heilbronn in the federal state of Baden–Wuerttemberg, Germany on 8 June 2022 and was submitted the next day to CVUA Stuttgart for post-mortem and microbiological investigations.

### 2.2. Bacteriological Examinations

The lungs, liver, spleen, kidneys, and the small and large intestines of the beavers were submitted to bacteriological examinations. Aseptic cut surfaces of the organs were directly streaked onto the surface of sheep blood agar and selective MacConkey or Gassner agar (intestines), and subsequently incubated at 37 °C under atmospheric conditions for two days. In addition, all abscesses were sampled and material streaked on sheep blood agar and selective MacConkey agar. Grown colonies were taken directly from the agar plates and pure cultures were prepared for identification and characterization.

### 2.3. Identification and Characterization

*Corynebacterium* cultures were identified at the species level by MALDI-TOF MS (matrix-assisted laser desorption/ionization—time of flight mass spectrometry; Bruker Daltonics, Bremen, Germany) using an extended reference database [31]. The isolates were further compared by spectroscopic analysis using Fourier-transform infrared-spectroscopy (FT-IR; Bruker Optics, Ettlingen, Germany) according to Martel et al. [32].

Metabolic profiles were created using the GEN III OmniLog^®^ ID System (Biolog, Hayward, CA, USA) according to the manufacturer’s instructions. The type strain *C. ulcerans* DSM 46325^T^ (NCTC 7910^T^) was used for quality control and reference.

Phospholipase D activity was detected by streaking the *C. ulcerans* isolates at right angles to a centrally growing *Staphylococcus* (*S*.) *aureus* (WDCM 00034) and *Rhodococcus* (*R*.) *equi* (ATCC 33701) by typical inhibition of the staphylococcal and enhancement of the rhodococcal hemolysis, respectively [33].

In addition, these results were verified by partial sequencing of the *rpoB*-gene according to the recommendations of Khamis et al. [34], using the primers C2700F and C3130R for identification of the *Corynebacterium* spp. In addition, sequencing of the 16-23S internal transcribed spacer region (16-23S ITS) was performed using modified broad-range primers based on sequences provided by Johnson et al. [35] and Hunt et al. [36]: 1522F_mod TGCGGYTGGAWCACCTCCTT, 189R_mod TACTDAGATGTTTCAVTTC. Sequence data were evaluated by comparison with sequence entries in the GenBank using BLASTN [37].

Detection of the diphtheria toxin gene was performed by real-time PCR [38], and toxin production was verified by the Elek immunoprecipitation assay following the optimized modified protocol by Melnikov et al. [39].

### 2.4. Antimicrobial Susceptibility Testing

Antimicrobial susceptibility testing of the *C. ulcerans* isolates was performed by the broth microdilution method for determination of minimum inhibitory concentrations (MIC) to 14 different antimicrobials. The tests were carried out in commercially available antimicrobial microdilution plates for companion animals following the manufacturer’s instructions (Sifin Diagnostics, Berlin, Germany, according to guidelines of the German Veterinary Society (DVG) research group on antimicrobial resistance). The following 14 antimicrobials were tested (ranges given in mg/L): amoxicillin/clavulanic acid (0.063/0.031–16/8), ampicillin (0.125–8), cefovecin (0.25–4), cephalexin (0.5–16), chloramphenicol (1–16), clindamycin (0.031–1), enrofloxacin (0.016–2), erythromycin (0.023–4), gentamicin (0.063–4), oxacillin (0.063–2), penicillin G (0.063–4), pradofloxacin (0.004–1), tetracycline (0.063–8) and trimethoprim/sulfamethoxazole (0.25/4.75–2/38). Results (MIC values) of the antimicrobial susceptibility testing were interpreted using clinical breakpoints according to EUCAST [40] for broth microdilution testing. The type strain *C. ulcerans* DSM 46325^T^ (NCTC 7910^T^) was used for quality control and reference.

### 2.5. Whole Genome Sequencing (WGS) Analysis

WGS of the three isolates was performed with Illumina paired-end sequencing (Illumina, San Diego, CA, USA) after DNA isolation using the Promega Maxwell system and library preparation with the Illumina DNA prep kit. WGS read data were checked for quality and absence of contamination with Illumina SAV software, fastqc [41] and kraken2 [42] before conducting in-depth analysis.

For genetic species confirmation, genomes were trimmed [43] and assembled with spades [44]. After performing assembly QC via QUAST [45], an average nucleotide identity (ANI) analysis was carried out with the tool PyANI (Application and Python module for whole-genome classification of microbes using Average Nucleotide Identity, Version v0.2, of the Leighton Pritchard Strathclyde Institute for Pharmacy and Biomedical Sciences University, Glasgow, Scotland) based on blast algorithm, as described in [23]. This involved pairwise comparison of the assemblies with public genomes of *C. diphtheriae* NCTC 11397^T^, *C. belfantii* FRC0043^T^, *C. rouxii* FRC0190^T^, *C. pseudotuberculosis* DSM 20689^T^ (ATCC 19410^T^), *C. silvaticum* DSM 109166^T^ (KL 0182^T^, CVUAS 4292^T^), *C. ulcerans* DSM 46325 ^T^ (NCTC 7910^T^) and *C. epidermidicanis* DSM 45586^T^, as a closely related outgroup of the *C. diphtheriae* group.

Multi-locus sequence typing (MLST) and core genome (cg) MLST were performed in Ridom SeqSphere+ (Ridom GmbH, Münster, Germany). MLST was conducted based on the seven genes *atpA, dnaE, dnaK, fusA, leuA, odhA, rpoB* described in [46,47]. Minimum spanning trees (MST) were constructed based on cgMLST allelic profiles of the sequenced *C. ulcerans* isolates, ignoring missing alleles during the pairwise profile comparisons and using two different ad hoc typing schemes:

(1)A pan-genomic *C. ulcerans*/*C. pseudotuberculosis* scheme with 193 species-overlapping target loci, as previously described in [23]. The addition of isolate typing profiles for *C. pseudotuberculosis* and *C. silvaticum* enables differentiation of the most closely related species.(2)A *C. ulcerans* cgMLST scheme of 1211 target loci, previously described by Berger et al. [25], with the addition of other animal-based *C. ulcerans* isolate typing profiles.

## 3. Results

### 3.1. Post-Mortem and Bacteriological Examinations

#### 3.1.1. Beaver 1 (County of Main–Spessart, Bavaria; 21 September 2015)

The beaver was a young male (without head, tail and skin) with a remaining body weight of 1.0 kg. The animal was cachectic and showed low-grade autolysis. Post-mortem examination of the inner organs revealed no special findings. Abscesses could not be found.

Bacteriological examination revealed a moderate growth (about 100 colonies) of *C. ulcerans* (151012433-002) and *Staphylococcus aureus* in the lungs and a low growth (about 10 colonies) of *Yersinia enterocolitica* in the colon.

The cause of the cachexia and death of this animal could not be identified.

#### 3.1.2. Beaver 2 (County of Soest, North Rhine–Westphalia, 7 April 2021)

The second beaver was an old female with a body weight of 13.6 kg. This animal was also cachectic with atrophy of the coronary fat and pronounced edema were detected in the subcutaneous tissue of the ventral thoracic region and abdomen. Furthermore, post-mortem also revealed swollen lymph nodes and scars in the throat region. An abscess was detected in the subcutis in the region of the left hip and a further abscess at the third left rib fused with the left lung (Figure 1). Swab samples from these abscesses were taken for bacteriological examinations. The lungs showed a severe edema and black-colored areas in the tissue. Histologically, the lung revealed a multifocal granulomatous pneumonia with conidia demarcated in connective tissue typical of adiaspiromycosis, caused by the fungus *Emmonsia crescens*. Bacteriological testing showed severe growth (about 1000 colonies) of *Streptococcus* (*S*.) *castoreus* in the lungs, liver and abscess, accompanied by moderate growth (about 100 colonies) of *C. ulcerans* (S 477/6/21). The age, the bite wounds and the infections of the animal are to be considered as the cause of cachexia and death.

#### 3.1.3. Beaver 3 (District of Heilbronn, Baden-Wuerttemberg, 8 June 2022)

The third beaver was a young male with a body weight of 16 kg. The nutritional condition of the beaver was poor, resulting in a loss of the coronary fat. The external examination of the body revealed previous injuries in the throat area and multiple abscesses in the lymph nodes localized in the inguinal region and on the cranial upper jaw. Swab samples from the lymph node abscesses and the inguinal region were taken for bacteriological examination. Black colored stipples in the tissue were verified by histo-pathological examinations as multiple granuloma-enclosing conidia that are typical of adiaspiromycosis (*Emmonsia crescens*). Bacteriological examinations showed severe growth (about 1000 colonies) of *C. ulcerans* (CVUAS 33950), accompanied by a moderate growth (about 100 colonies) of *S. castoreus* in the specimens taken from the abscesses. The injuries and the infections are considered the cause of the cachexia and subsequent death of the beaver.

### 3.2. Identification and Characterization of the Corynebacterium sp.

The three *C. ulcerans* isolates were unequivocally identifiable at species level by MALDI-TOF MS (Score 2.3–2.5) (metadata and single MALDI-TOF mass-spectra of the isolates used in this study are listed in the MALDI-UP catalogue [https://maldi-up.ua-bw.de] and are accessible on request) and FT-IR spectroscopy (Figure 2), as well as by partial sequencing of the 16-23S rRNA intergenic spacer region (NCBI BLAST: 100% query cover, differences of percent identity of 3.3–3.8% and 7.3–7.5% to the closely related species *C. pseudotuberculosis* and *C. silvaticum*, respectively). Evaluation of the partial *rpoB* gene sequences using NCBI BLAST revealed high similarity between *C. ulcerans* and *C. silvaticum* (NCBI BLAST: 100% query cover, differences of percent identities to *C. silvaticum* of 0.0–0.3%). However, a differentiation from *C. pseudotuberculosis* was unequivocal (100% query cover, difference of percent identity of 4.9%).

All of the three *C. ulcerans* isolates from the beavers produced phospholipase D. This could be demonstrated on sheep blood agar by inhibition of the hemolysis of *S. aureus* and a synergistic hemolysis with *R. equi*. These *C. ulcerans* isolates are classified as toxigenic since they carry and express the diphtheria toxin gene.

### 3.3. Biochemical Investigations

A broad analysis of biochemical identification properties was assessed for the three isolates under study, for which results are shown in Figure 3 and Appendix A. As reference strain *C. ulcerans* DSM 46325^T^ (NCTC 7910^T^) was used. Differences in biochemical reactions between the beaver isolates and the reference strain become apparent. Despite not all members of the *C. diphtheriae* group having been included in the Omnilog database, the three *C. ulcerans* isolates and the reference strain *C. ulcerans* DSM 46325^T^ (NCTC 7910^T^) were clearly identified at the species level by the Omnilog software, with a distance value of ≥0.500 (DIST 5.439–5.597).

Analysis of metabolic activities revealed positive results for 16 out of 94 biochemical reactions (Figure 3). The results did not differ between the tested isolates and the type strain for *C. ulcerans* DSM 46325^T^ (NCTC 7910^T^) (Figure 3, red line). However, slight differences in metabolic activities could be detected. In seven reactions (N-acetyl-D-glucosamine, fusidic acid, troleandomycin, rifamycin SV, minocycline, lincomycin, and tetrazolium violet), isolate S 477/6/21 (Beaver 2) showed a slightly higher metabolic rate compared to the other isolates. Contrarily, isolate CVUAS 33950 (Beaver 3) metabolized pectin at a slightly higher activity level (Appendix A).

### 3.4. Antimicrobial Susceptibility Testing

The results of antimicrobial susceptibility testing are listed in Table 1. Consistently, all isolates were resistant to clindamycin (MIC > 2 mg/L) and susceptible to all other antimicrobials tested. No differences in resistance patterns were evident compared with the type strain DSM 46325^T^ (NCTC 7910^T^).

However, when using the newly recommended EUCAST breakpoints for erythromycin (MIC > 0.06 mg/L), only isolate S 477/6/21 (Beaver 2) displayed a resistant phenotype (0.064 mg/L) [40], while type strain DSM 46325^T^ (NCTC 7910^T^) was susceptible.

### 3.5. Whole Genome Sequencing (WGS) Analysis

WGS was carried out on the three *C. ulcerans* isolates and the resulting data were used for confirmation of the genetic species by ANI and for phylogenetic clustering by cgMLST, using two different schemes. ANI analysis in comparison with type strain genomes from the *C. diphtheriae* group confirmed the results of the MALDI-TOF MS, FT-IR and 16-23S ITS analyses. The three isolates were clearly classified as *C. ulcerans,* with ANI values of 98% compared with the *C. ulcerans*-type strain. These ANI values are above the threshold value of ~95–96% for taxonomic delineation of prokaryotic species [48], whereas ANI values for all other species were below 91% (Table 2).

MLST analysis based on seven housekeeping loci classified all three isolates as *C. ulcerans* sequence type (ST)- 332 (*atp*A: 42, *dnaE*: 33, *dnaK*: 78, *fusA*: 49, *leuA*: 48, *odhA*: 43, *rpoB*: 40).

CgMLST analysis was performed with two different ad hoc schemes to be able to classify the isolates from two different perspectives. For the generation of the pangenomic scheme 193 overlapping genes from *C. ulcerans* and *C. pseudotuberculosis* were used. These divide in the resulting species-specific MST branches, with the beavers’ isolates grouping in the branch of the species *C. ulcerans* but being separate in different branches from *C. pseudotuberculosis*, as well as from the closely related *C. silvaticum*. For comparison, genomic profiles from other animal-derived isolates of the three species, generated with the same scheme, were included in the analysis. The beaver-derived isolates clearly group on the branch of *C. ulcerans* (Figure 4). Thereby, the species classification can be clearly confirmed.

Using the cgMLST scheme, including 1211 core genes from *C. ulcerans*, generated and described in [25], the resulting MST shows the genetic relationship of the beaver-derived *C. ulcerans* isolates to each other and to other genomic profiles from animal-derived *C. ulcerans* isolates, added for comparison. The beavers’ isolates show thereby the highest similarity to each other with allelic differences (AD) of 14–26 and to the nearest relative *C. ulcerans* from another host species derived from a dog quantified with an AD of 86 (Figure 5).

## 4. Discussion

*C. ulcerans* occupies a special status among the corynebacteria belonging to the *C. diphtheriae* species complex due to its potential production of the diphtheria toxin, its broad spectrum of hosts ranging from livestock and companion animals to wild animals, and its pronounced zoonotic character [26,49,50]. The wide spectrum of susceptible animals has been comprehensively presented by Berger et al. [25]. *C. ulcerans* is considered to be an emerging human pathogen causing cases of diphtheria-like illness, which have outnumbered diphtheria cases caused by *C. diphtheriae* in Europe for the last two decades [26,27,28,31,32]. Of particular concern has become the transmission of *C. ulcerans* from companion animals, especially cats and dogs, to humans. Such infections can be severe with even lethal outcomes, which have been documented in numerous case reports and summarized in reviews [26,49]. In contrast, human-to-human transmission of *C. ulcerans* plays only a minor role for this pronounced zoonotic pathogen [51]. In parallel, cases of infections with toxigenic *C. ulcerans* displaying clinical signs of the respiratory tract and ulcerative skin lesions have recently been reported in dogs and cats [52]. Furthermore, hunting dogs are reported to represent a potential link for transmission of *C. ulcerans* from wild animals to humans and companion animals through direct contact with wildlife [53]. *C. ulcerans* infections have also been observed in a greater number of hedgehogs, who often live in close proximity to humans [25,32]. The increasing population of beavers in Germany might form a novel reservoir for *C. ulcerans*. However, no cases of *C. ulcerans* in other European countries have so far been reported. Therefore, the prevalence of *C. ulcerans* in beavers is generally unknown and thus the present study is only able to give an indication of the presence of pathogenic *C. ulcerans* in beavers. The role of established and new toxigenic and NTTB strains of animal and human origin has been recently and comprehensively evaluated by Prygiel et al. [21]. The close relationship between animal and human pathogenic corynebacteria isolates suggests that other animal pathogenic isolates may also be pathogenic to humans [24]. Thus, it is of great interest, from the One Health point of view, whether beaver populations represent a novel reservoir for pathogenic, zoonotic *C. ulcerans*. The three deceased beavers included in this study were spatio-temporally found widely separated from each other in the German federal states of Bavaria, North Rhine–Westphalia and Baden–Wuerttemberg, in which Eurasian beavers had been reintroduced and reestablished [54]. Although the source of toxigenic *C. ulcerans* in beavers remains hidden, beavers have to be considered as a further, previously undetected free-ranging species posing a reservoir for this zoonotic agent. Taking this into account, monitoring of wildlife on pathogenic bacteria and molecular epidemiology should be implemented as an indispensable tool of the One Health approach.

Detection of *C. ulcerans* in the lungs of Beaver 1 was not associated with major pathological changes, suggesting a colonization of the lungs as commensal, as is also described for the respiratory tract and oral cavity of dogs [55,56] and cats [57,58]. However, since this animal was submitted decapitated, we cannot rule out bacterial spread to this organ. Conversely, the other two beavers developed multiple abscesses in the subcutis and lymph nodes of the trunk and showed scarring of the skin in the throat area. These injuries are indicative of bite wounds resulting from ranking or territorial fights as also reported for other animal species like otters [59], water rats [60], squirrels [61] or hedgehogs in which *C. ulcerans* had also been detected [25,32]. In addition to these wildlife species, beavers also serve as a so-far-undiscovered new host. Thus, transmission of *C. ulcerans* from beavers to other semi-aquatic rodents like the invasive muskrats and nutrias living in the same freshwater habitats is conceivable. However, the role of beavers as a carrier and possible reservoir for *C. ulcerans* among semi-aquatic mammals currently remains speculative. Therefore, semi-aquatic wildlife should also be consistently monitored on pathogenic *Corynbacterium* spp.

Identification of corynebacteria at the species level in routine bacteriological laboratories is difficult because similar profiles can be retrieved by conventional phenotypical and even current test methods such as MALDI-TOF MS [52,62,63]. However, the GEN III OmniLog system, a fully automated system for the biochemical identification of microorganisms, unequivocally confirmed the placement of isolates to *C. ulcerans*. Although the OmniLog system does not always clearly identify coryneforms [64], the isolates tested and the reference strain were clearly identified as *C. ulcerans*. The same result was also reported by Berger et al. [25]. Despite minor differences, isolates from beavers represented a largely homogenous population within the variability range of *C. ulcerans.* The Omnilog software was additionally used to compare the metabolic curves of each of the 94 metabolic reactions per isolate and in comparison to the reference strain. For this purpose, the metabolic profile curves of the measured individual reactions were overlaid and the differences visually highlighted (Figure 3). These results show that the metabolic responses in terms of negative and positive reactions do not differ between the tested isolates and the reference strain. These results are consistent with those previously reported [25,65] (Appendix A). However, it can be seen that the isolate from Beaver 2 appears to be slightly more metabolically active than the other isolates or the reference strain. Interestingly, this is the same isolate that was exclusively resistant to erythromycin according to the current EUCAST recommendations on breakpoints. Susceptibility testing revealed sensitivity to erythromycin, except for one isolate (S 477/6/21, Beaver 2). In contrast, all isolates were revealed to be resistant to clindamycin. This antimicrobial susceptibility pattern agrees with previous studies, which report antimicrobial susceptibility to erythromycin for the majority of human and animal *C. ulcerans* isolates tested. In contrast, susceptibility to clindamycin has been reported from sensitive to intermediary to resistant [25,66,67].

The quality of MALDI-TOF MS results strongly relies on comprehensive and sophisticated databases representing all currently established *Corynebacterium* spp. of the *C. diphtheriae* group [23,31]. This is of even more importance, as closely related corynebacteria, too, must be clearly differentiated due to their different zoonotic potential [23,31,52,62]. In this context, the recent separation of isolates from the so-called wild boar cluster of *C. ulcerans* as the novel species *C. silvaticum* has to be considered [23]. This *Corynebacterium* sp. is most closely related to *C. ulcerans* based on 16S rRNA gene and *rpoB* gene sequences; thus, *C. silvaticum* strains were formerly classified as *C. ulcerans* [23,31]. In our study, however, *C. ulcerans* could unequivocally be differentiated from *C. silvaticum* using MALDI-TOF MS and FT-IR, supported by an extended spectral database, as described in the catalogue of the MALDI-User Platform (https://MALDI-UP.ua-bw.de (accessed on 13 June 2022)) [68]. Using extended and comprehensive spectral databases for MALDI-TOF MS and FT-IR analysis, precise and reproducible identification of the current corynebacteria can be carried out quickly and accurately [23,31,69]. The same applies to 16-23S ITS sequences, which allowed a clear identification of *Corynebacterium* spp. in this study.

Production of virulence factors, e.g., phospholipase D (PLD), presence of the *tox* gene, and production of the diphtheria toxin were verified for all three *C. ulcerans* isolates as recommended [60,62]. Whereas PLD is reportedly produced by all pathogenic *C. ulcerans* isolates [26,52], production of the diphtheria toxin, which is considered a main virulence factor [26,70], was found in only about 5% of the *C. diphtheria* and in 50–60% of the *C. ulcerans* isolates [28]. A recent study in France revealed a striking frequency in toxigenic isolates from dogs and cats among companion animals [71]. Strains of toxigenic as well as the NTTB phenotype in *C. ulcerans* in wild and exotic animal species have already been found in otters [69], water rats [60], squirrels [61] and hedgehogs [25,32]. These animal species have been reported to suffer from suppurative to necrosuppurative, ulcerative, superficial to deep skin lesions, and infestation of various inner organs like the lung and heart, irrespective of the production of the diphtheria toxin.

In addition to *C. ulcerans*, *S. castoreus* was isolated in abscesses from two animals and also, in one of these animals, in the liver and lungs. *S. castoreus* has been described as a separate, beta-hemolytic *Streptococcus* sp. belonging to the group A streptococci, which have previously been isolated from multiple bite wounds in a beaver [72]. There are few reports on the detection of *S. castoreus* exclusively in beavers, considering that this *Streptococcus* sp. is strictly host-specific [73,74]. In these reports, *S. castoreus* has also been isolated from suppurative lesions in association with bite wounds and systemic infections, as well as from apparently healthy beavers. However, *S. castoreus* has been assessed as being an opportunistic pathogen in beavers, primarily colonizing the normal microbiota of the oral, respiratory and genital mucosa. Finally, tissue lesions and concomitant colonization by opportunistic pathogenic bacteria such as *Actinomyces*, *Corynebacterium* or streptococci and Gram-negative obligate anaerobic bacteria might lead to severe poly-microbial suppurative inflammation and abscesses [74,75,76,77].

Overall, the number of beavers that are available for mircobiological investigations is very limited because impairment of beavers in their natural habitat is strictly prohibited. Thus, active monitoring on the health of beavers is not realizable, meaning that data are available from only a limited number of free-ranging beavers found dead. In the federal states of Hesse and Baden–Wuerttemberg, 17 and 115 animals have been subjected to post-mortem examinations in the last 10 years, respectively. However, in the federal state of North Rhine–Westphalia, this was the first beaver examined by post-mortem analysis.

Whole-genome sequencing has become a valuable tool for the molecular characterization of bacterial isolates as a basis of molecular epidemiology [78,79]. Evaluation of the genome sequence data of the *C. ulcerans* isolates from the three beavers revealed a genome size of approximately 2.5 Mbp and a GC content of 53.4%. This data is consistent with that previously reported for *C. ulcerans* isolates originating from animals and humans [80]. ANI and pangenomic cgMLST analysis clearly classify the isolates as *C. ulcerans,* which is consistent with MALDI-TOF MS, FT-IR and 16-23S ITS analysis.

WGS-derived MLST based on seven housekeeping loci allowed a classification of the beaver *C. ulcerans* isolates as ST-332. There are only a few reports on this sequence type, detected in hedgehogs [25], in a cat [47] and in humans [47,51,81,82]. Detection of ST-332 in animals and humans is an indication of a possible zoonotic transmission of this pathogen and emphasizes the importance of MLST and cgMLST for unravelling transmission routes of *C. ulcerans* [47]. A MLST based on *C. ulcerans* cgMLST analysis using 1211 core genes shows that the genetic distance between the *C. ulcerans* isolates from the beavers is smaller than that of the isolates from other animal host species. A similar cgMLST scheme in *C. diphtheriae* with a size range of 1500 alleles showed a comparable range of allelic distances like single nucleotide polymorphism (SNP) phylogenies; closely related isolates of *C. diphtheriae* and *C. ulcerans* showed distances of only a few alleles/SNPs, respectively [25,70,83]. Given this knowledge, we conclude that distances in this range (>20) for the beaver isolates are no indication of a close genetic relationship or even direct transmission.

## 5. Conclusions

Wildlife disease surveillance is an important instrument for the early detection of emerging diseases and zoonoses. Therefore, examinations of all wild animals found dead are a vital element of wildlife disease surveillance. In this context, we examined three Eurasian beavers and were able to detect *C. ulcerans* in this wild animal species for the first time. Based on WGS data, the isolates show close genetic but no clonal relationship to each other or to isolates originating from other host species. *C. ulcerans* isolates are known as zoonotic agents. In this respect, these isolates from beavers must be considered as potential zoonotic candidates, and beavers should be recognized as a heretofore unknown possible reservoir for pathogenic *C. ulcerans*. In order to cover the important and complex group of zoonotic corynebacteria, interdisciplinary cooperation with regard to the One Health approach is pivotal.

Further investigations on pathogenic *C. ulcerans* in wildlife supported by molecular epidemiology should be conducted to obtain deeper knowledge about the occurrence, epidemiology and impact of *C. ulcerans* in beavers and other semi-aquatic mammals, particularly considering the ever-increasing beaver population in Europe.

## Figures and Tables

**Figure 1 pathogens-12-00979-f001:**
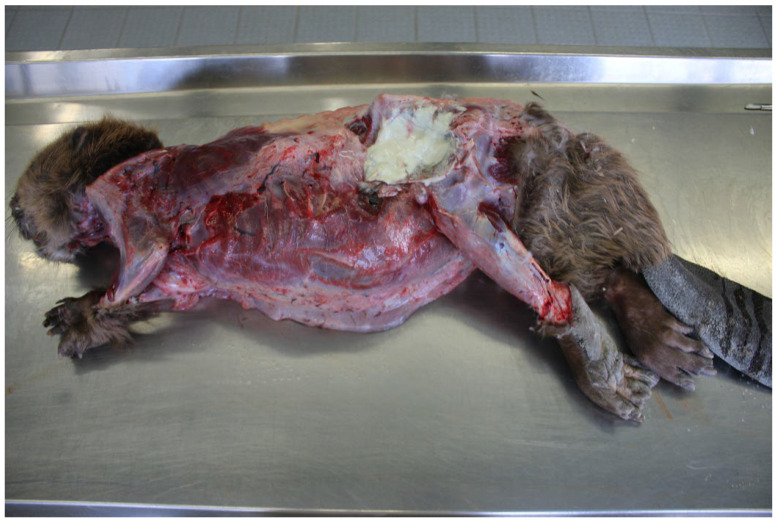
Carcass of a Eurasian beaver showing an abscess in the subcutis in the region of the left hip detected during post-mortem examination. *C. ulcerans* could be isolated from the pus of this abscess.

**Figure 2 pathogens-12-00979-f002:**
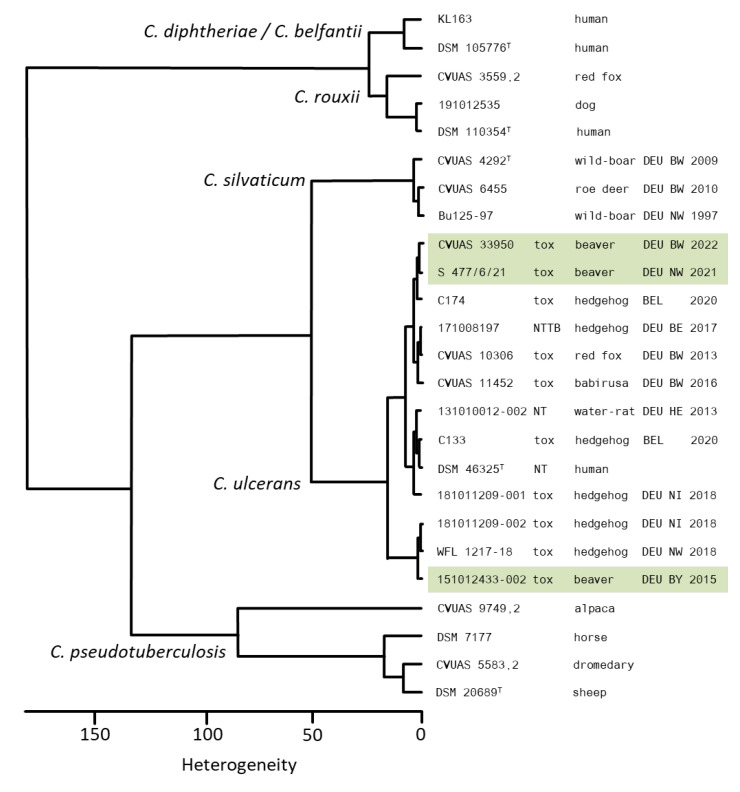
Dendrogram of IR spectra including three beaver *C. ulcerans* isolates, in comparison to a set of further *C. ulcerans* isolates from wildlife and zoo-animals, *C. diphtheriae*, *C. belfantii*, *C. rouxii*, *C. silvaticum* and *C. pseudotuberculosis*, including type strains (^T^). *C. ulcerans* strains: 151012433-002 = Beaver 1 S 477/6/21 = Beaver 2, CVUAS 33950 = Beaver 3.

**Figure 3 pathogens-12-00979-f003:**
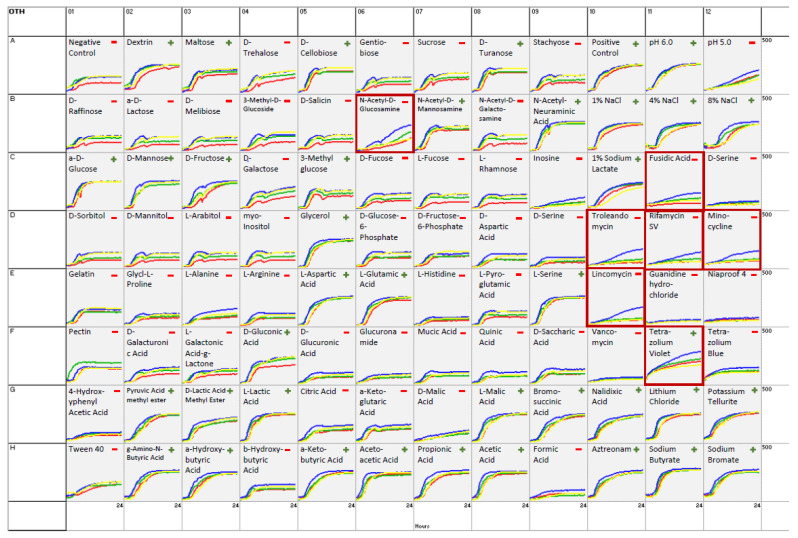
Metabolic GEN III OmniLog profiles of *C. ulcerans*: 151012433-002 (Beaver 1, yellow line); S 477/6/21 (Beaver 2, blue line); CVUAS 33950 (Beaver 3, green line); and DSM 46325^T^ (NCTC 7910^T^, reference strain, red line), incubated at 30 °C for 24 h. Visual representation of the kinetics in growth curves as measured from OD values for different substrates; +: positive reaction, −: negative reaction; well A1: negative control; well A10: positive control; red boxes highlighting special substrate utilization of S 477/6/21 in wells B06, C11, D10, D11, D12, E10, F11.

**Figure 4 pathogens-12-00979-f004:**
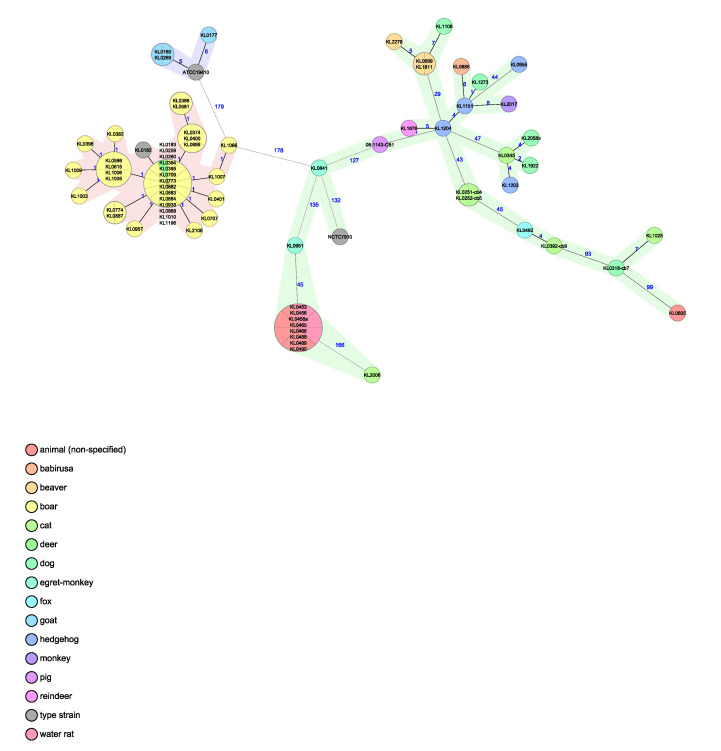
MST of the cgMLST analysis of the three beaver *C. ulcerans* isolates compared with isolates from *C. ulcerans*, *C. silvaticum* and *C. pseudotuberculosis*, using a pangenomic scheme with 193 target loci. Samples are color-coded according to their host species and branches are highlighted with color according to the *Corynebacterium* spp.: light green = *C. ulcerans*, light red = *C. silvaticum*, light blue = *C. pseudotuberculosis*. NCTC 7910^T^ = DSM 46325^T^ (*C. ulcerans*), KL 0182^T^ = DSM 109166^T^, CVUAS 4292^T^ = (*C. silvaticum*) [23], ATCC 19410^T^ = DSM 20689^T^ (*C. pseudotuberculosis*); C. *ulcerans* strains: KL 0689 = 151012433-002 (Beaver 1), KL 1811 = S 477/6/21 (Beaver 2), KL 2278 = CVUAS 33950 (Beaver 3).

**Figure 5 pathogens-12-00979-f005:**
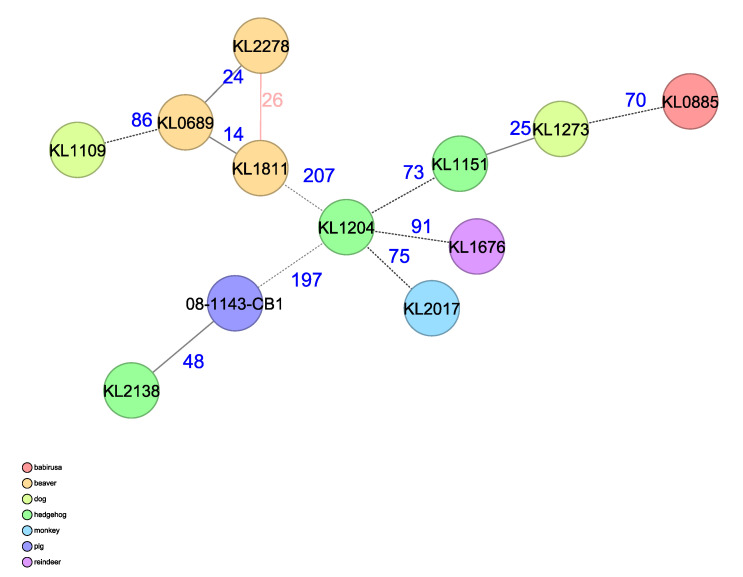
MST of the cgMLST analysis of the three beaver *C. ulcerans* isolates in comparison to isolates from *C. ulcerans*, using a species-specific cgMLST scheme with 1211 target loci. Allelic differences are indicated and samples are color-coded according to their host species. *C. ulcerans* strains: KL 0689 = 151012433-002 (Beaver 1), KL 1811 = S 477/6/21 (Beaver 2), KL 2278 = CVUAS 33950 (Beaver 3).

**Table 1 pathogens-12-00979-t001:** Minimal inhibitory concentration (MIC) values (mg/L) for the three beaver *C. ulcerans* isolates and the type strain DSM 46325^T^ (NCTC 7910^T^) determined by the broth microdilution method using microtiter plates.

Antimicrobials	MIC Range	MIC Results of *C. ulcerans* Isolates			
		Beaver 1 151012433-002	Beaver 2 S 477/6/21	Beaver 3 CVUAS 33950	DSM 46325^T^ (NCTC 7910^T^)
AMC	0.063/0.031–16/8	≤0.063/0.031	≤0.063/0.031	≤0.063/0.031	≤0.063/0.031
AMP	0.125–8	≤0.125	≤0.125	≤0.125	≤0.125
CFV	0.25–4	≤0.25	≤0.25	≤0.25	≤0.25
CEX	0.5–16	≤0.5	≤0.5	≤0.5	≤0.5
CMP	1–16	≤1	≤1	≤1	≤1
CLI	0.031–1	>2	>2	>2	>2
ENR	0.016–2	=0.063	=0.063	=0.031	=0.031
ERY	0.023–4	=0.047	=0.064	=0.023	=0.047
GEN	0.063–4	=2	=4	=4	=4
OXA	0.063–2	=1	=1	=1	=1
PEN	0.063–4	≤0.063	≤0.063	≤0.063	≤0.063
PRX	0.004–1	=0.031	=0.016	=0.008	=0.008
TET	0.063–8	=0.25	=0.25	=0.25	=0.25
T/S	0.25/4.75–2/38	=0.5/9.5	=0.5/9.5	≤0.25/4.25	≤0.25/4.25

AMC: amoxicillin/clavulanic acid; AMP: ampicillin; CFV: cefovecin; CEX: cephalexin; CMP: chloramphenicol; CLI: clindamycin; ENR: enrofloxacin; ERY: erythromycin; GEN: gentamicin; OXA: oxacillin; PEN: penicillin G; PRX: pradofloxacin; TET: tetracycline; T/S: trimethoprim/sulfamethoxazole.

**Table 2 pathogens-12-00979-t002:** Average nucleotide identity (ANI) values of WGS assemblies of the three beaver *C. ulcerans* isolates compared to public-type strain genomes.

Type Strain Genome	Species	Beaver 1 151012433-002	Beaver 2 S 477/6/21	Beaver 3 CVUA S33950
DSM 46325^T (^NCTC 7910^T^)	*C. ulcerans*	0.988	0.987	0.987
KL 0182^T^ (DSM 109166 ^T^, CVUAS 4292 ^T^)	*C. silvaticum*	0.904	0.902	0.904
ATCC 19410^T^ (DSM 20689^T^)	*C. pseudotuberculosis*	0.845	0.844	0.845
NCTC 11397^T^	*C. diphtheriae*	0.742	0.742	0.742
FRC 0190^T^	*C. rouxii*	0.741	0.741	0.740
FRC 0043^T^	*C. belfantii*	0.740	0.739	0.739
DSM 45586^T^	*C. epidermidicanis*	0.732	0.732	0.732

## Data Availability

WGS raw data are available at the short reads archive of the National Center of Biotechnology Information (NCBI) under bioproject PRJNA938404 (with the following biosample accessions for beaver 1: SAMN33434792, beaver 2: SAMN33434793, beaver 3: SAMN33434794).

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
