# Peer review of "Corynebacterium ulcerans Infections in Eurasian Beavers (Castor fiber)"

_pathogens, 2023, doi:10.3390/pathogens12080979_

Round 1

Reviewer 1 Report

The study “Corynebacterium ulcerans Infections in Eurasian Beavers (Cas-2 tor fiber)” is important and novel as it expands our knowledge of C. ulcerans infections in beavers, provides insights into disease pathology, utilizes advanced molecular techniques for characterization, highlights antibiotic resistance patterns, and enhances our understanding of the genetic relatedness of C. ulcerans strains. However, following points should be considered before accepting the study for publication.

In the abstract, it would be helpful to briefly mention the implications or significance of these findings in terms of public health or wildlife conservation.

In the methodology, there are several shortcomings e.g., the lack of information on sample size (how much sample from one carcass) and the absence of a clear sampling strategy. The specific criteria for selecting the examined beavers are not provided, which raises questions about the representativeness of the findings. Additionally, the description of bacteriological examinations is limited, leaving out important details such as the number of colonies examined and criteria for selection.

In the results section, following changes are required. While cachexia is considered the cause of death in the first beaver, the underlying factors leading to cachexia are not fully explored or identified. This leaves gaps in understanding the primary drivers of the observed symptoms. Secondly, the bacteriological examination identifies the presence of C. ulcerans, Staphylococcus aureus, and Yersinia enterocolitica, but additional information on their virulence, antimicrobial resistance, and potential interactions is lacking, which will be essential for the clarity of the results. Additionally, biochemical investigations were conducted, the results are not extensively discussed or compared with reference strains, limiting the interpretation of metabolic activities and their relevance to disease development. Furthermore, the antimicrobial susceptibility results focus only on clindamycin and erythromycin, leaving out other commonly used antibiotics. The lack of a broader antimicrobial susceptibility profile hinders the assessment of treatment options.

The discussion section of the provided text lacks certain elements that are essential for a comprehensive and critical analysis. Firstly, it does not thoroughly address the limitations of the study. Without acknowledging the study's shortcomings, it becomes challenging to evaluate the reliability and generalizability of the findings. Additionally, the discussion does not compare the results with previous studies or provide a broader context for the research. This omission hinders the ability to assess the novelty and significance of the findings and limits the understanding of how the study contributes to the existing knowledge in the field. Moreover, the discussion lacks specific recommendations for future research directions, which would help guide further investigations and provide a roadmap for advancing knowledge in the field. Overall, the discussion section could be improved by incorporating these crucial elements to enhance the overall quality and impact of the study.

There are some formatting error in the write-up. The abbreviations should be written in the first exposure i.e MLST should be written in full form in the abstract section at Line; 22. Secondly, at Page 2 Line 48, full stop is missing after reference.

 Good but minor editing of the English language is required.

Author Response

Reviewer 1

Comments and Suggestions for Authors

The study “Corynebacterium ulcerans Infections in Eurasian Beavers (Cas-2 tor fiber)” is important and novel as it expands our knowledge of C. ulcerans infections in beavers, provides insights into disease pathology, utilizes advanced molecular techniques for characterization, highlights antibiotic resistance patterns, and enhances our understanding of the genetic relatedness of C. ulcerans strains. However, following points should be considered before accepting the study for publication.

In the abstract, it would be helpful to briefly mention the implications or significance of these findings in terms of public health or wildlife conservation.

Authors‘ comment: This aspect has been considered in the chapter Abstract.

In the methodology, there are several shortcomings e.g., the lack of information on sample size (how much sample from one carcass) and the absence of a clear sampling strategy. The specific criteria for selecting the examined beavers are not provided, which raises questions about the representativeness of the findings. Additionally, the description of bacteriological examinations is limited, leaving out important details such as the number of colonies examined and criteria for selection.

Authors‘ comment: The missing data has been supplemented in the chapters Material & Methods and Results. The three beavers had not been selected. These animals were sent dead to our institutes.

In the results section, following changes are required. While cachexia is considered the cause of death in the first beaver, the underlying factors leading to cachexia are not fully explored or identified. This leaves gaps in understanding the primary drivers of the observed symptoms.

Authors‘ comment: It is true that the cause of the cachexia and death of the first beaver could not conclusively be determined. This has been adjusted.

Secondly, the bacteriological examination identifies the presence of C. ulcerans, Staphylococcus aureus, and Yersinia enterocolitica, but additional information on their virulence, antimicrobial resistance, and potential interactions is lacking, which will be essential for the clarity of the results.

Authors‘ comment (Bevaer1): We agree with Reviewer 1 that a holistic analysis of all potential pathogens would have been desirable. Unfortunately, respective isolates have not been stored. Because the principal scope of our study was to highlight the clustering and phenotypic as well as genotypic properties of C. ulcerans isolates, we hope that Reviewer 1 will agree to omit these requested data.

Additionally, biochemical investigations were conducted, the results are not extensively discussed or compared with reference strains, limiting the interpretation of metabolic activities and their relevance to disease development. Furthermore, the antimicrobial susceptibility results focus only on clindamycin and erythromycin, leaving out other commonly used antibiotics. The lack of a broader antimicrobial susceptibility profile hinders the assessment of treatment options.

Authors‘ comment: We thank the reviewer for this objection. The Omnilog was used primarily for phenotypic determination and as another method to identify the C. ulcerans isolates. Of course, a reference strain was included in this procedure. The missing reference strain information has been added to the method section, accordingly. The Omnilog software can additionally be used to compare the metabolic curves of the individual 94 metabolic reactions per isolate and the reference strain. This has been done in Figure 3. These results show that the metabolic responses in terms of negative and positive do not differ between the tested isolates and to the reference strain. However, it can be seen that the isolate from Beaver 2 appears to be slightly more metabolically active than the other isolates or the reference strain. Within the scope of this study, it is not possible to assess what cause or effect this might have. However, this gives reason to look at this in more detail in the context of further studies.  The missing discussion of the biochemical results has been added.

Thank you very much for your comment on antimicrobial testing. The aim of the antimicrobial testing was to compare the resistance profile of the C. ulcerans isolates among each other and to the reference strain. It was not intended to serve results for therapy recommendation. The test was performed by microbouillon dilution using standard veterinary plate layouts for pet animals, including 14 antimicrobial substance. Information for the reference strain used was added. The results have been discussed in depth and comparison with the literature has been extended.

The discussion section of the provided text lacks certain elements that are essential for a comprehensive and critical analysis.

Firstly, it does not thoroughly address the limitations of the study. Without acknowledging the study's shortcomings, it becomes challenging to evaluate the reliability and generalizability of the findings.

Authors‘ comment: The limitations of the present study have been considered in the chapter Discussion.

Additionally, the discussion does not compare the results with previous studies or provide a broader context for the research. This omission hinders the ability to assess the novelty and significance of the findings and limits the understanding of how the study contributes to the existing knowledge in the field.

Authors‘ comment: The findings of C. ulcerans in other wildlife species and in beavers serving as a possible new reservoir for C. ulcerans has been considered.

Moreover, the discussion lacks specific recommendations for future research directions, which would help guide further investigations and provide a roadmap for advancing knowledge in the field. Overall, the discussion section could be improved by incorporating these crucial elements to enhance the overall quality and impact of the study.

Authors‘ comment: Recommendations for further research have been integrated.

There are some formatting error in the write-up. The abbreviations should be written in the first exposure i.e. MLST should be written in full form in the abstract section at Line; 22. Secondly, at Page 2 Line 48, full stop is missing after reference.

Authors‘ comment: The errors have been corrected.

Comments on the Quality of English Language

Good but minor editing of the English language is required.

Authors‘ comment: The manuscript has been revised in this regard.

Reviewer 2 Report

These authors presented a report of three cases of C. ulcerans infections in Eurasian Beavers. Although it is indeed an interesting and complete report from a microbiology and antibiotic resistance perspective, I do have some concerns regarding this manuscript. The way it is written and the way the results are interpreted should be clearly improved before publication. There are also some inconsistencies that I mention in the current report.

Major concerns

In general, the way this manuscript is written should be improved, to be more concise and scientific, and less descriptive. Authors use long sentences, and repetitive words, making the sentences harder to read and follow. Just as an illustration L171-174 could be abbreviated to "This animal was also cachetic. Furthermore, it also revealed swollen lymph nodes and scars in the throat region."

Moreover, the authors barely discuss their results from a One Health perspective, even though I believe your work is very related to the One Health principles. They introduced the concept and mentioned it again in the Conclusions but I believe a true interpretation and comparison with other studies should be made in the Discussion section. For instance, authors should try to provide some answers to the following questions: "Why finding these pathogens or these antibiotic resistance patterns is relevant for the conservation of these species (or other species) or to the humans that live nearby?" OR "Do these findings (and their consequences) only affect beavers? What are other species from the same habitats should also be taken into consideration? OR "Are beavers good sentinels for infectious diseases or antibiotic resistance? Why? What others can be used in future assessments and how?"

Other specific concerns (some are major as well):

1) The first part of the abstract is repetitive and disorganized. Pleae rephrase it. This sentence  L14-15 "During examinations, Corynebacterium (C.) ulcerans were able to be isolated in beavers for the first time." should be gone and the name of the bacterium species should also be written in italics. 

2) Some keywords are sentences (and maybe too specific) rather than true "keywords". Keywords should be expressions with no more than 3 words.

3) L35-39: authors should provide the current status of the species (after repopulation) in Europe, referring to its conservation status and general distribution. Please check the IUCN website.

4) L 54: "species" can be replaced by "spp." . Consider this for the rest of the manuscript. When you refer to more than one species of the same genus (as Corynebacterium), you should use "spp." . When only one is considered "sp." should be used.

5) L54: "belonging" . I am not an English native speaker but I believe this is not the correct verb form here. Please revise your manuscript for English writing. 

6) L164-165 "The beaver was a young male (without head, tail and skin) with a remaining body weight of 1.0 kg. The animal was cachectic and showed low-grade autolysis". For me, this is very strange and the first thing that comes to me is: how are you sure it is a beaver since almost 95% of the animal is missing (considering the weight of beaver number 3)? This concerns me. This is a very small body part. I don't know if this should be included and analysed with beavers number 2 or 3, but perhaps I am not seeing the full picture. I would really appreciate it if the authors can help me understand.

7) Figure 1 should be presented closer to the paragraph where you mentioned it. I believe it is very far away.

Suggestion:

I believe the article structure (Introduction, Methodology, Results,...) makes it harder to read and follow the three case reports. I think the authors should follow Case Report structure (please see "Instructions for Authors") and present an Introduction, then describe the case reports (animal location and post-mortem presentation/details). Then, authors can refer to the sample collection and the results of each lab analysis (and its methods). At the end, they can provide a global discussion of the three case reports together. Otherwise, I believe it is more difficult to follow like it is, especially between the methods and results. But this is merely a suggestion that I would like to leave for your consideration.

I have nothing else to add and I wish the authors everything good with their work.

Author Response

Reviewer 2

Comments and Suggestions for Authors

These authors presented a report of three cases of C. ulcerans infections in Eurasian Beavers. Although it is indeed an interesting and complete report from a microbiology and antibiotic resistance perspective, I do have some concerns regarding this manuscript. The way it is written and the way the results are interpreted should be clearly improved before publication. There are also some inconsistencies that I mention in the current report.

Major concerns

In general, the way this manuscript is written should be improved, to be more concise and scientific, and less descriptive. Authors use long sentences, and repetitive words, making the sentences harder to read and follow. Just as an illustration L171-174 could be abbreviated to "This animal was also cachectic. Furthermore, it also revealed swollen lymph nodes and scars in the throat region."

Authors‘ comment: Thank you for the advice. We agree that some of the sentences might be too long and have tried to shorten several of them. The manuscript has been revised in this regard.

Moreover, the authors barely discuss their results from a One Health perspective, even though I believe your work is very related to the One Health principles. They introduced the concept and mentioned it again in the Conclusions but I believe a true interpretation and comparison with other studies should be made in the Discussion section. For instance, authors should try to provide some answers to the following questions: "Why finding these pathogens or these antibiotic resistance patterns is relevant for the conservation of these species (or other species) or to the humans that live nearby?" OR "Do these findings (and their consequences) only affect beavers? What are other species from the same habitats should also be taken into consideration? OR "Are beavers good sentinels for infectious diseases or antibiotic resistance? Why? What others can be used in future assessments and how?"

Authors‘ comment: These aspects have been considered in the chapters Abstract and Discussion. The results of antibiotic susceptibility testing have been discussed in more depth.

Other specific concerns (some are major as well):

1) The first part of the abstract is repetitive and disorganized. Please rephrase it. This sentence L14-15 "During examinations, Corynebacterium (C.) ulcerans were able to be isolated in beavers for the first time." should be gone and the name of the bacterium species should also be written in italics.

Authors‘ comment: This chapter has been reorganized. Results of molecular analysis are now presented coherently in the text. The sentence in the lines 14-15 has been removed.

2) Some keywords are sentences (and maybe too specific) rather than true "keywords". Keywords should be expressions with no more than 3 words.

Authors‘ comment: The keywords have been revised.

3) L35-39: authors should provide the current status of the species (after repopulation) in Europe, referring to its conservation status and general distribution. Please check the IUCN website.

Authors‘ comment: The current status of the Eurasian Beaver according to the IUCN Red List of Threatened Species has been considered.

4) L 54: "species" can be replaced by "spp." . Consider this for the rest of the manuscript. When you refer to more than one species of the same genus (as Corynebacterium), you should use "spp." . When only one is considered "sp." should be used.

Authors‘ comment: „species“ has been replaced by „sp.“ or „spp.“, respectively,  in context with bacterial genera.

5) L54: "belonging". I am not an English native speaker but I believe this is not the correct verb form here. Please revise your manuscript for English writing.

Authors‘ comment: „belonging“ should be correct. The manuscript had been revised by a native speaker prior to submission and the word „belonging“ has been confirmed as correct in this context.

6) L164-165 "The beaver was a young male (without head, tail and skin) with a remaining body weight of 1.0 kg. The animal was cachectic and showed low-grade autolysis". For me, this is very strange and the first thing that comes to me is: how are you sure it is a beaver since almost 95% of the animal is missing (considering the weight of beaver number 3)? This concerns me. This is a very small body part. I don't know if this should be included and analyzed with beaver number 2 or 3, but perhaps I am not seeing the full picture. I would really appreciate it if the authors can help me understand.

Authors‘ comment (Beaver 1): Wildlife authorities have submitted beaver 1 to our lab after the given parts were removed for unknown reasons. However, together with the mentioned species diagnosis there was no doubt that the available body parts were consistent with the species diagnosis of a Eurasian beaver. From our perspective, it makes sense to include this animal, because restricted necropsy and biological examinations of all inner organs of the torso could be performed.

7) Figure 1 should be presented closer to the paragraph where you mentioned it. I believe it is very far away.

Authors‘ comment: Figure 1 has been moved behind the results of the post mortem examinations.

Suggestion:

I believe the article structure (Introduction, Methodology, Results,...) makes it harder to read and follow the three case reports. I think the authors should follow Case Report structure (please see "Instructions for Authors") and present an Introduction, then describe the case reports (animal location and post-mortem presentation/details). Then, authors can refer to the sample collection and the results of each lab analysis (and its methods). At the end, they can provide a global discussion of the three case reports together. Otherwise, I believe it is more difficult to follow like it is, especially between the methods and results. But this is merely a suggestion that I would like to leave for your consideration.

Authors‘ comment: We agree that information on the beavers‘ origin and date of discovery of the animals is missing in the chapter „Results“. However, we think that including the methodology in the chapter „Results“ would overload this chapter. Therefore, we added information on the origin and date of discovery of the beavers behind the headline of Beaver 1, 2 and 3.

Round 2

Reviewer 2 Report

The authors have addressed most of my comments regarding this manuscript, and I absolutely recognize this. 

However, there are still some aspects to be pointed it regarding this manuscript:

The authors clearly say that the manuscript was revised by an English native speaker. However, there are still some language and punctuation inconsistencies. For example, there is certainly a comma or full stop missing here, otherwise the too-long sentence does not make any sense:

L15-16: "Since wildlife is an important source of zoonotic infectious diseases monitoring of invasive and reintroduced species is crucial with respect to the One Health approach."

Moreover, it is still very strange to me what happened to beaver number 1 and if its analysis gives a positive contribution to your work or if it is actually prejudicial. I have already given my opinion regarding this, and I do not want to insist too much on this point. 

The rest of the manuscript, especially the discussion section has improved significantly and this should be recognized.

Author Response

Journal: Pathogens (ISSN 2076-0817)

Manuscript ID: pathogens-2466298

Type: Article

Title: Corynebacterium ulcerans Infections in Eurasian Beavers (Castor fiber)

Authors: Reinhard Sting, Catharina Pölzelbauer , Tobias Eisenberg , Rebecca Bonke , Birgit Blazey , Martin Peters , Karin Riße , Andreas Sing , Anja Berger , Alexandra Dangel , Jörg Rau

Section: Emerging Pathogens

Special Issue: Emerging Zoonoses: The Bridge between Human and Animal Diseases in the One Health Paradigm

Submission Date: 07 June 2023

Date of submission of the reviewers‘ comments: 5 July 2023

Date of resubmission: 15 July 2023

Reviewer 2

Comments and Suggestions for Authors

These authors presented a report of three cases of C. ulcerans infections in Eurasian Beavers. Although it is indeed an interesting and complete report from a microbiology and antibiotic resistance perspective, I do have some concerns regarding this manuscript. The way it is written and the way the results are interpreted should be clearly improved before publication. There are also some inconsistencies that I mention in the current report.

Major concerns

In general, the way this manuscript is written should be improved, to be more concise and scientific, and less descriptive. Authors use long sentences, and repetitive words, making the sentences harder to read and follow. Just as an illustration L171-174 could be abbreviated to "This animal was also cachectic. Furthermore, it also revealed swollen lymph nodes and scars in the throat region."

Authors‘ comment: Thank you for the advice. We agree that some of the sentences might be too long and have tried to shorten several of them. The manuscript has been revised in this regard. A native speaker proofread the manuscript after revision.

Moreover, the authors barely discuss their results from a One Health perspective, even though I believe your work is very related to the One Health principles. They introduced the concept and mentioned it again in the Conclusions but I believe a true interpretation and comparison with other studies should be made in the Discussion section. For instance, authors should try to provide some answers to the following questions: "Why finding these pathogens or these antibiotic resistance patterns is relevant for the conservation of these species (or other species) or to the humans that live nearby?" OR "Do these findings (and their consequences) only affect beavers? What are other species from the same habitats should also be taken into consideration? OR "Are beavers good sentinels for infectious diseases or antibiotic resistance? Why? What others can be used in future assessments and how?"

Authors‘ comment: These aspects have been considered in the chapters Abstract and Discussion. The results of antibiotic susceptibility testing have been discussed in more depth.

Other specific concerns (some are major as well):

1) The first part of the abstract is repetitive and disorganized. Please rephrase it. This sentence L14-15 "During examinations, Corynebacterium (C.) ulcerans were able to be isolated in beavers for the first time." should be gone and the name of the bacterium species should also be written in italics.

Authors‘ comment: This chapter has been reorganized. Results of molecular analysis are now presented coherently in the text. The sentence in the lines 14-15 has been removed.

2) Some keywords are sentences (and maybe too specific) rather than true "keywords". Keywords should be expressions with no more than 3 words.

Authors‘ comment: The keywords have been revised.

3) L35-39: authors should provide the current status of the species (after repopulation) in Europe, referring to its conservation status and general distribution. Please check the IUCN website.

Authors‘ comment: The current status of the Eurasian Beaver according to the IUCN Red List of Threatened Species has been considered.

4) L 54: "species" can be replaced by "spp." . Consider this for the rest of the manuscript. When you refer to more than one species of the same genus (as Corynebacterium), you should use "spp." . When only one is considered "sp." should be used.

Authors‘ comment: „species“ has been replaced by „sp.“ or „spp.“, respectively,  in context with bacterial genera.

5) L54: "belonging". I am not an English native speaker but I believe this is not the correct verb form here. Please revise your manuscript for English writing.

Authors‘ comment: „belonging“ should be correct. The manuscript had been revised by a native speaker prior to submission and again after our revision. The word „belonging“ has been confirmed as correct in this context.

6) L164-165 "The beaver was a young male (without head, tail and skin) with a remaining body weight of 1.0 kg. The animal was cachectic and showed low-grade autolysis". For me, this is very strange and the first thing that comes to me is: how are you sure it is a beaver since almost 95% of the animal is missing (considering the weight of beaver number 3)? This concerns me. This is a very small body part. I don't know if this should be included and analyzed with beaver number 2 or 3, but perhaps I am not seeing the full picture. I would really appreciate it if the authors can help me understand.

Authors‘ comment (Beaver 1): Wildlife authorities have submitted beaver 1 to our lab after the given parts were removed for unknown reasons. However, together with the mentioned species diagnosis there was no doubt that the available body parts were consistent with the species diagnosis of a Eurasian beaver. From our perspective, it makes sense to include this animal, because restricted necropsy and biological examinations of all inner organs of the torso could be performed.

7) Figure 1 should be presented closer to the paragraph where you mentioned it. I believe it is very far away.

Authors‘ comment: Figure 1 has been moved behind the results of the post mortem examinations.

Suggestion:

I believe the article structure (Introduction, Methodology, Results,...) makes it harder to read and follow the three case reports. I think the authors should follow Case Report structure (please see "Instructions for Authors") and present an Introduction, then describe the case reports (animal location and post-mortem presentation/details). Then, authors can refer to the sample collection and the results of each lab analysis (and its methods). At the end, they can provide a global discussion of the three case reports together. Otherwise, I believe it is more difficult to follow like it is, especially between the methods and results. But this is merely a suggestion that I would like to leave for your consideration.

Authors‘ comment: We agree that information on the beavers‘ origin and date of discovery of the animals is missing in the chapter „Results“. However, we think that including the methodology in the chapter „Results“ would overload this chapter. Therefore, we added information on the origin and date of discovery of the beavers behind the headline of Beaver 1, 2 and 3.